# A mixed-methods systematic review of post-viral fatigue interventions: Are there lessons for long Covid?

**Sally Fowler-Davis**[1]*, **Katharine Platts**[1], **Michael Thelwell**[1], **Amie Woodward**[2], **Deborah Harrop**[1†]

**1** Advanced Wellbeing Research Centre, Health Research Institute, Sheffield Hallam University, Sheffield, United Kingdom, **2** Department of Health Sciences, University of York, York, United Kingdom

† Deceased.
* s.fowler-davis@shu.ac.uk

**Data Availability Statement:** All relevant data are within the paper and its Supporting Information files.

## Abstract

### Objectives

Fatigue syndromes have been widely observed following post-viral infection and are being recognised because of Covid19. Interventions used to treat and manage fatigue have been widely researched and this study aims to synthesise the literature associated with fatigue interventions to investigate the outcomes that may be applicable to 'long Covid'.

### Method

The study was registered with PROSPERO (CRD42020214209) in October 2020 and five electronic databases were searched. Papers were screened, critically appraised and data extracted from studies that reported outcomes of fatigue interventions for post-viral syndromes. The narrative synthesis includes statistical analysis associated with effectiveness and then identifies the characteristics of the interventions, including identification of transferable learning for the treatment of fatigue in long Covid. An expert panel supported critical appraisal and data synthesis.

### Results

Over 7,000 research papers revealed a diverse range of interventions and fatigue outcome measures. Forty papers were selected for data extraction after final screening. The effectiveness of all interventions was assessed according to mean differences (MD) in measured fatigue severity between each experimental group and a control following the intervention, as well as standardised mean differences as an overall measure of effect size. Analyses identified a range of effects–from most effective MD -39.0 [95% CI -51.8 to -26.2] to least effective MD 42.28 [95% CI 33.23 to 51.34]–across a range of interventions implemented with people suffering varying levels of fatigue severity. Interventions were multimodal with a range of supportive therapeutic methods and varied in intensity and requirements of the participants. Those in western medical systems tended to be based on self- management and education principles (i.e., group cognitive behavioural therapy (CBT).

**Funding:** SFD received part funding from Westfield Health https://www.westfieldhealth.com/my-westfield/overview for the purpose of a research assistant (namely AW and MT) only, all other contributions were undertaken under the auspices of the employment contract of the University. The funders had no role in study design, data collection and analysis, decision to publish, or preparation of the manuscript.

**Competing interests:** The authors have declared that no competing interests exist.

## Conclusion

Findings suggest that the research is highly focussed on a narrow participant demographic and relatively few methods are effective in managing fatigue symptoms. Selected literature reported complex interventions using self-rating fatigue scales that report effect. Synthesis suggests that long Covid fatigue management may be beneficial when a) physical and psychological support, is delivered in groups where people can plan their functional response to fatigue; and b) where strengthening rather than endurance is used to prevent deconditioning; and c) where fatigue is regarded in the context of an individual's lifestyle and home-based activities are used.

## Introduction

Fatigue is a common problem resulting from a range of conditions including cancer, multiple sclerosis (MS), and post-viral syndromes. Fatigue symptoms are always limiting but if there is an identifiable cause (e.g., poor sleep patterns) and lasts for less than three months, can be managed by resolving the underlying cause [1]. Chronic fatigue is experienced with a greater intensity and longer duration, causing severe disruption to daily life, functional activity and quality of life [2, 3]. The debilitating and prolonged nature of fatigue can pose significant economic consequences for society and significant anxiety and depression for the individual.

Fatigue management in practice has been recommended for cancer [4] and multiple-sclerosis [5], in rheumatoid arthritis [6, 7] and in a workplace context [8] and is most notably the core component of treatment for Chronic Fatigue Syndrome/ Myalgic Encephalomyelitis (CFS/ME) and fibromyalgia. Complex interventions can include exercise, psychological/behavioural, dietary and 'alternative' medicine including acupuncture and yoga. Interventions for fatigue can be home-based or undertaken in a clinical setting and studies of multi-modal interventions, i.e. crossover trials or studies involving a pharma component, as a complex intervention, have proliferated in recent years and are the subject of several Cochrane Reviews [9, 10].

In 2007, the National Institute of Care Excellence (NICE) identified the effectiveness of rest and management approaches for fatigue experience in the context of CFS/ME [11] recommending cognitive behavioural therapy (CBT) and graded exercise therapy (GET). These methods were substantiated by a programme of research focussed on results of evidence from several randomised controlled trials in the UK [12–14]. However, new guidance drafted in 2020 and currently under further development (NICE Guidance: GID-NG10091) [15] cited a "lack of evidence for the effectiveness of these interventions" based on a wide-spread reaction from patients who recognised the potentially harmful effects of over-generalised advice on exercise. NICE recommendations were deemed to be problematic because the focus is exclusively on Grading of Recommendations Assessment, Development and Evaluation (GRADE) standards to evaluate the evidence. Further work to identify fatigue interventions for use in holistic rehabilitation therefore remains an important challenge [16].

The recent Covid 19 pandemic has resulted in the early reporting of the later effects of the viral infection that can be characterised as a post-viral syndrome [17]. People with post-Covid Syndrome (popularly known as 'long Covid') suffers can experience a long-term state of chronic fatigue characterised by post-exertional exhaustion [18]. This appears to have a common aetiology to CFS/ME type fatigue, but long Covid fatigue co-exist with other symptoms including shortness of breath and associated anxiety [19]. The persistence of fatigue, at around

three months post-infection [20] appears to be associated with moderate to severe depression [21] and a worsened quality of life [19]. It appears that hospitalisation is not necessarily predictive of long Covid, and Covid-19 not predictive of outcomes [22] and there is a non-specific sequalae of long Covid [23].

Although a substantial body of studies on short-term outcomes of Covid-19 inpatients has already been produced, the literature associated with long-term outcomes is limited [24]. Whilst many individuals recover and have resumed their usual functional activity, a residual number of people require follow up and a small literature is building around the response from health and care services, including a call for an integrated team-based response [25]. With the development of NHS long Covid Clinics [26] there was an early recognition that post-viral fatigue was a common symptom, with patients reporting significant functional limitations for many months after their initial infection [27]. The World Health Organisation (WHO) has called for recognition, research, and rehabilitation in line with the growing demand for treatment and management [28].

The aim of this study is to systematically review the literature associated with fatigue management interventions, their characteristics and outcomes, and to identify the characteristics of treatments that may be useful in the management of long Covid.

## Materials and methods

The present review was conducted in accordance with preferred reporting for systematic review guidelines (PRISMA). A review protocol was developed and registered with the NIHR International Prospective Register of Systematic Reviews (PROSPERO), registration number CRD42020214209.

### Eligibility criteria

A Population, Intervention, Comparator, Outcome, Study-type (PICOS) framework was used to define the eligibility criteria. The population of interest was characterised as people of any age across all countries and of all ethnicities, based on post-viral or bacterial infection syndromes that manifest as post-viral fatigue. Those undergoing radiation or chemotherapy, or those with fatigue caused by auto-immune response following cancer or MS were excluded. Those with physiological or acute fatigue associated with fever, infection, sleep disturbances, pregnancy, extreme physical activity (excessive energy consumption), work-related burnout, or a primary depression/psychosis diagnosis were excluded on the basis that fatigue was not experienced because of past viral infection.

Fatigue management or energy conservation interventions were associated with exercise, psychological, behavioural, dietary, diet supplements or complementary/alternative medicine (CAM) interventions. Those included had reported outcome measures and were undertaken in a non-inpatient setting. Pharmacological-only interventions were excluded. Mixed interventions and studies involving a pharmacological component were included only when fatigue outcome measurement data could be extracted.

The principal outcome of interest was fatigue, and only studies that assessed fatigue as a distinct and separate outcome attributable to a fatigue management intervention were selected. Both physical and mental fatigue were included in the definition as measured by a recognised fatigue research measure. Secondary outcomes of interest were acceptability of intervention to users, mechanisms of action, and intervention characteristics including duration, intervention type, timing of intervention, mode of delivery and by whom (professional/peer), group nature, and infrastructure e.g., technology assisted.

Primary research from the period 2002–2020 was deemed eligible for inclusion, to allow for literature related to the SARS (SARS-CoV-1) epidemic of 2002–2004 to be captured. Qualitative, quantitative and mixed-methods designs were included. Studies comparing effectiveness of interventions against usual care groups or non-exposed control groups were selected, while uncontrolled trials were excluded. Reviews and case studies were excluded. Non-English language literature was excluded due to the financial costs associated with translation.

## Information sources

A search of online databases (CENTRAL, CINAHL, MEDLINE (EBSCO), ProQuest (APA PsycINFO), SCOPUS, SportDISCUS) was conducted to identify relevant literature in October 2020. Where documents were not available directly from the publisher's website, access was requested via Sheffield Hallam University document supply services. The International Clinical Trials Registry Platform (World Health Organization) was searched, as well as the UK Clinical Trials Gateway (NHS, National Institute for Health Research).

## Search strategy

The search strategy comprised two facets: terms to describe fatigue or causes of post-viral fatigue; combined with terms to describe types of interventions, e.g., clinical trials. The Boolean operators AND and OR were used to combine search terms. Controlled vocabulary terms were used where available, and when supported by the database. Refer to S1 File for a copy of the search strategy as written up for MEDLINE (EBSCO) and ProQuest (APA PsycInfo). The search strategy was adapted for other databases.

## Study selection and quality appraisal

Following the removal of duplicates, the titles and abstracts of all studies yielded from the literature searches were screened by a review author to assess suitability for inclusion against PICOS criteria. A second reviewer independently screened 10% of the studies by title and abstract. Papers selected for full-text screening were retrieved and independently assessed for eligibility by review authors (MT, KP and SFD), with 10% of the papers screened by a second review author (DH) to check for conformity of selection and to quality assess the level of agreement between reviewers. Disagreements in screening decisions were resolved through discussions within the research team. All papers selected at final full text screening stage were subject to a preliminary quality appraisal using the Mixed Methods Appraisal Tool (MMAT) (Table 1). For each study, an overall quality score was calculated based on the number of MMAT criteria that each study satisfied. Papers were not automatically excluded based on outcomes from the MMAT and additional quality appraisal was undertaken continuously by review authors. Trustworthiness of the selected papers was undertaken in consultation with the expert panel who shared their thoughtful and informative but necessarily more subjective views of the literature, and a mechanistic approach was not used [29].

## Data extraction

Papers were selected for final inclusion where they met all criteria and data was harvested from the selected papers and collated in an a priori data extraction form created for the purposes of the review. The data extraction exercise was piloted by all review authors with three authors undertaking the data extraction exercise.

Extracted data included: bibliographic information, country of study, hypotheses/research question, intervention setting, study design, recruitment and sampling procedures, participant

**Table 1. Overview of studies selected for review.**

| Lead author (Year) (A-Z) | Title | Intervention groups (n) | Participants<br>Age: Mean (SD)<br>Sex: Female / Male (%) | | Setting | Study design | Intervention duration | MMAT Quality Score |
|---|---|---|---|---|---|---|---|---|
| Clark (2017) | Guided graded exercise self-help plus specialist medical care versus specialist medical care alone for chronic fatigue syndrome (GETSET): a pragmatic randomised controlled trial | SMC (102)<br>GET & SMC (97) | 38.4 (11.9) years | 79% / 21% | Secondary care / Specialist clinic | Open-label, pragmatic RCT | 8 weeks | 5 |
| Dailey (2013) | Transcutaneous electrical nerve stimulation reduces pain, fatigue and hyperalgesia while restoring central inhibition in primary fibromyalgia | No TENS (41)<br>Active TENS (41)<br>Placebo TENS (41) | 49.2 (12) years | 98% / 2% | Not stated | Randomised, placebo-controlled cross-over | 3 weeks | 5 |
| Demirbag (2012) | The effects of sleep and touch therapy on symptoms of fibromyalgia and depression | Control (54)<br>TMA (54)<br>SMA (54) | 42.5 (11.8) years | 110% / 0% | Community / Clinic | Randomised comparison study with control | 6 weeks | 3 |
| El Mokadem (2020) | Three principles/innate health: The efficacy of psycho-spiritual mental health education for people with chronic fatigue syndrome | Waiting list (11)<br>Three Principles/ Innate Health (11) | 42.9 (10.5) (range: 19–66) years | 86% / 14% | Not stated | Randomised trial with waitlist control | 8 weeks | 4 |
| Ericsson (2016) | Resistance exercise improves physical fatigue in women with fibromyalgia: a randomized controlled trial | Relaxation (63)<br>Group-based GET (67) | 51.4 (9.4) years | 100% / 0% | Community / Clinic | Multicentre RCT | 15 weeks | 3 |
| Fernie (2016) | Treatment Outcome and Metacognitive Change in CBT and GET for Chronic Fatigue Syndrome | CBT (116)<br>GET (55) | 40.8 (12.5) (range: 18–75) years | | Secondary care / outpatient | Comparison of CBT & GET in practice | 4 months | 3 |
| Fitzgibbon (2018) | Evidence for the improvement of fatigue in fibromyalgia: a 4-week left dorsolateral prefrontal cortex repetitive transcranial magnetic stimulation randomized-controlled trial | Sham rTMS (12)<br>rTMS (14) | 45.6 (12.8) years | 92% / 8% | Secondary care / Lab-based | Randomised, double-blind placebo controlled | 4 weeks | 5 |
| Friedberg (2016) | Efficacy of two delivery modes of behavioural self-management in severe chronic fatigue syndrome | SMC (46)<br>Self-management & web diaries (39)<br>Self-management & paper diaries (39) | 48.4 (11.5) years | 88% / 12% | Primary care / home-based | Randomised comparison | 3 months | 4 |
| Hansen (2013) | Heart rate variability and fatigue in patients with chronic fatigue syndrome after a comprehensive cognitive behaviour group therapy program | Healthy controls (21)<br>CBT / GET / Group therapy (19) | 41.6 (range: 29–67) years | 100% / 0% | Not stated | Comparison (healthy controls vs. CFS patients) | 4 days | 4 |
| Heald (2019) | Service based comparison of group cognitive behaviour therapy to waiting list control for chronic fatigue syndrome with regard to symptom reduction and positive psychological dimensions | Waiting list (28)<br>Group CBT (28) | 43.1 (13.2) years | 61% / 39% | Primary care / specialist clinic | Repeated measures– individuals act as own control | 8 weeks | 4 |
| Jason (2010) | Provision of social support to individuals with chronic fatigue syndrome | Waiting list (15)<br>Social support (15) | 57.6 (13) years | 83% / 17% | Primary care / home-based | Randomised trial with waitlist control | 4 months | 4 |
| Keijmel (2017) | Effectiveness of Long-term Doxycycline Treatment and Cognitive-Behavioral Therapy on Fatigue Severity in Patients with Q Fever Fatigue Syndrome (Qure Study): a Randomized Controlled Trial | Placebo (52)<br>Doxycycline (52)<br>CBT (50) | 43.8 (12.1) years | 48% / 52% | Secondary care / Lab-based | RCT | 24 weeks | 4 |

*(Continued)*

**Table 1.** (Continued)

| Lead author (Year) (A-Z) | Title | Intervention groups (n) | Participants Age: Mean (SD) Sex: Female / Male (%) | Setting | Study design | Intervention duration | MMAT Quality Score |
|---|---|---|---|---|---|---|---|
| Kim (2013) | Indirect moxibustion (CV4 and CV8) ameliorates chronic fatigue: a randomized, double-blind, controlled study | Sham Moxibustion (20) | Median: 44 (range: 32–63) years | Lab-based | Double-blinded RCT | 4 weeks | 4 |
| | | Moxibustion (25) | 78% / 22% | | | | |
| Kim (2015) | Acupuncture for chronic fatigue syndrome and idiopathic chronic fatigue: a multicenter, nonblinded, randomized controlled trial | SMC (50) | 42.2 (11.7) years | Secondary care | RCT | 4 weeks | 4 |
| | | Acupuncture & SMC (49) | 65% / 35% | | | | |
| | | Sa-am Acupuncture & SMC (51) | | | | | |
| Lee (2015) | The effect of oriental medicine music therapy on idiopathic chronic fatigue | Waiting list (15) | 44.9 (12.4) years | Secondary care / outpatient | Randomised trial with waitlist control | 2 weeks | 4 |
| | | Oriental medicine music therapy (15) | 90% / 10% | | | | |
| Maddali (2016) | Efficacy of rehabilitation with Tai Ji Quan in an Italian cohort of patients with Fibromyalgia Syndrome | Educational course (22) | 52.2 (12.2) years | Primary care / home-based | Randomised comparison | 16 weeks | 4 |
| | | Tai Ji Quan (22) | | | | | |
| Marques (2015) | Effects of a Self-regulation Based Physical Activity Program (The '4-STEPS') for Unexplained Chronic Fatigue: a Randomized Controlled Trial | SMC (46) | 48.1 (11) years | Community / home-based | Multicentre RCT | 12 weeks | 3 |
| | | Self-regulation Based Physical Activity (45) | 98% / 2% | | | | |
| Marques (2017) | Efficacy of a randomized controlled self-regulation based physical activity intervention for chronic fatigue: mediation effects of physical activity progress and self-regulation skills | SMC (46) | 48.1 (11) years | Primary care / home-based | Multicentre RCT | 12 weeks | 3 |
| | | Self-regulation Based Physical Activity (45) | 98% / 2% | | | | |
| Mist (2018) | Randomized Controlled Trial of Acupuncture for Women with Fibromyalgia: group Acupuncture with Traditional Chinese Medicine Diagnosis-Based Point Selection | Group Education (14) | 54 (12.4) years | Community / Specialist clinic | Random allocation / repeated measures | 10 weeks | 4 |
| | | Group Acupuncture (16) | 100% / 0% | | | | |
| Ng (2013) | Acupuncture for chronic fatigue syndrome: a randomized, sham-controlled trial with single-blinded design | Sham Acupuncture (49) | 40.9 (6.6) years | Lab-based | Single-blinded RCT | 4 weeks | 3 |
| | | Acupuncture (50) | 69% / 31% | | | | |
| O'Dowd (2006) | Cognitive behavioural therapy in chronic fatigue syndrome: a randomised controlled trial of an outpatient group programme | SMC (51) | 41.1 (11.9) years | Secondary care / outpatient | Double-blind RCT | 16 weeks | 4 |
| | | CBT (52) | | | | | |
| | | GET (50) | 66% / 33% | | | | |
| Oka (2014) | Isometric yoga improves the fatigue and pain of patients with chronic fatigue syndrome who are resistant to conventional therapy: a randomized, controlled trial | Pharma (15) | 38.6 (12.6) years | Community / Clinic | RCT | 2 months | 4 |
| | | Yoga & Pharma (15) | 80% / 20% | | | | |
| Perrin (2011) | Muscle fatigue in chronic fatigue syndrome/myalgic encephalomyelitis (CFS/ME) and its response to a manual therapeutic approach: A pilot study | Healthy controls (9) | 35.8 (range: 20–55) years | Lab-based | Comparison–osteopathy vs self-selected treatment vs healthy control | 12 months | 3 |
| | | Osteopathic treatment (9) | | | | | |
| | | Any treatment (9) | 44% / 56% | | | | |

(*Continued*)

**Table 1.** (Continued)

| Lead author (Year) (A-Z) | Title | Intervention groups (n) | Participants | | Setting | Study design | Intervention duration | MMAT Quality Score |
|---|---|---|---|---|---|---|---|---|
| | | | Age: Mean (SD) | | | | | |
| | | | Sex: Female / Male (%) | | | | | |
| Powell (2004) | Patient education to encourage graded exercise in chronic fatigue syndrome: 2-year follow-up of randomised controlled trial | SMC (34) | 33.2 (10.3) years | | Outpatient / home-based | RCT | 12 months | 3 |
| | | GET: | | | | | | |
| | | Min. intervention (37) | 75% / 25% | | | | | |
| | | Telephone intervention (39) | | | | | | |
| | | Max. intervention (33) | | | | | | |
| Racine (2019) | Operant Learning Versus Energy Conservation Activity Pacing Treatments in a Sample of Patients With Fibromyalgia Syndrome: a Pilot Randomized Controlled Trial | Control (43) | No details | | Community / Clinic | RCT | 10 weeks | 3 |
| | | Operant learning (17) | | | | | | |
| | | Energy conservation activity pacing (24) | | | | | | |
| Raijmakers (2019) | Long-term effect of cognitive behavioural therapy and doxycycline treatment for patients with Q fever fatigue syndrome: One-year follow-up of the Qure study | Placebo (52) | 43.8 (12.1) years | | Secondary care / outpatient | RCT | 24 weeks | 4 |
| | | Doxycycline (52) | | | | | | |
| | | CBT (50) | 48% / 52% | | | | | |
| Ridsdale (2012) | The effect of counselling, graded exercise and usual care for people with chronic fatigue in primary care: a randomized trial | SMC & CBT booklet (75) | 39.8 (range: 34–46) years | | Primary care / home-based | RCT | 12 months | 5 |
| | | GET (71) | 78% / 22% | | | | | |
| | | Counselling (76) | | | | | | |
| Sharpe (2015) | Rehabilitative treatments for chronic fatigue syndrome: long-term follow-up from the PACE trial | SMC (115) | 38 (12) years | | Primary care / Community | Multicentre randomised trial | 6 months | 3 |
| | | APT & SMC (120) | 77% / 23% | | | | | |
| | | CBT & SMC (119) | | | | | | |
| | | GET & SMC (127) | | | | | | |
| Shu (2016) | Acupuncture and Moxibustion have Different Effects on Fatigue by Regulating the Autonomic Nervous System: a Pilot Controlled Clinical Trial | Healthy controls (15) | 37.1 (14.3) years | | Lab-based | RCT | 3 weeks | 3 |
| | | Acupuncture (15) | 73% / 27% | | | | | |
| | | Moxibustion (15) | | | | | | |
| Stubhaug (2008) | Cognitive-behavioural therapy v. mirtazapine for chronic fatigue and neurasthenia: randomised placebo-controlled trial | Placebo (24) | 46.32 (8.75) years | | Not stated | RCT | 24 weeks | 4 |
| | | Mirtazapine (25) | | | | | | |
| | | CBT (23) | 82% / 18% | | | | | |
| Tummers (2010) | Effectiveness of stepped care for chronic fatigue syndrome: a randomized noninferiority trial | SMC (85) | 38.1 (10.3) years | | Outpatient / home-based | Randomised non-inferiority study | 6 months | 2 |
| | | Self-instruction & CBT (84) | 79% / 21% | | | | | |
| UÄŸurlu (2017) | The effects of acupuncture versus sham acupuncture in the treatment of fibromyalgia: a randomized controlled clinical trial | Sham Acupuncture (25) | 45.4 (8.2) years | | Community / Specialist clinic | RCT | 5 weeks | 4 |
| | | Acupuncture (25) | 100% / 0% | | | | | |
| Van Hoof (2003) | Hyperbaric Therapy in Chronic Fatigue Syndrome | HBOT: | 42 (13) years | | Lab-based | Randomised sampling | 1 week | 5 |
| | | Healthy controls (13) | 66% / 34% | | | | | |
| | | Infection (13) | | | | | | |
| Vos-Vromans (2016) | Multidisciplinary rehabilitation treatment versus cognitive behavioural therapy for patients with chronic fatigue syndrome: a randomized controlled trial | CBT (60) | 40.3 (11.1) years | | Community | Multicentre RCT | 10 weeks | 3 |
| | | Multidisciplinary rehabilitation treatment (62) | 80% / 20% | | | | | |

(*Continued*)

**Table 1.** (Continued)

| Lead author (Year) (A-Z) | Title | Intervention groups (n) | Participants Age: Mean (SD) Sex: Female / Male (%) | Setting | Study design | Intervention duration | MMAT Quality Score |
|---|---|---|---|---|---|---|---|
| Wearden (2010) | Nurse led, home-based self-help treatment for patients in primary care with chronic fatigue syndrome: randomised controlled trial | SMC (100) Pragmatic rehab (95) Supportive listening (101) | 44.6 (11.4) years 78% / 22% | Primary care / home-based | Single-blind RCT | 18 weeks | 4 |
| Weatherley-Jones (2004) | A randomized, controlled, triple-blind trial of the efficacy of homeopathic treatment for chronic fatigue syndrome | Placebo (50) Homeopathic medication (53) | 38.9 (10.8) years 59% / 41% | Community / Clinic | Triple-blind RCT | 6 months | 4 |
| White (2011) | Comparison of adaptive pacing therapy, cognitive behaviour therapy, graded exercise therapy, and specialist medical care for chronic fatigue syndrome (PACE): a randomised trial | SMC (160) APT & SMC (159) CBT & SMC (161) GET & SMC (160) | 38 (12) years 77% / 23% | Secondary care | Multicentre randomised trial | 24 weeks | 4 |
| Wiborg (2015) | Randomised controlled trial of cognitive behaviour therapy delivered in groups of patients with chronic fatigue syndrome | Waiting list (68) CBT: Large group (68) Small group (68) | 37.9 (11.3) years 77% / 23% | Secondary care / outpatient | RCT | 6 months | 5 |
| Windhorst (2017) | Heart rate variability biofeedback therapy and graded exercise training in management of chronic fatigue syndrome: an exploratory pilot study | Biofeedback (13) GET (11) | 50.7 (9.3) years 100% / 0% | Secondary care / outpatient | Randomised study | 8 weeks | 5 |
| Wu (2020) | Observation on therapeutic efficacy of tuina plus cupping for chronic fatigue syndrome | Ginseng lozenges (50) Tuina & Cupping (50) | 40.1 (5.1) years 63% / 37% | Community / Specialist clinic | RCT | 16 weeks | 4 |

Cognitive behavioural therapy (CBT); Chronic Fatigue Syndrome (CFS); Graded Exercise Therapy (GET); Randomised controlled trial (RCT); Adaptive Pacing Therapy (APT); Standard Medical Care (SMC); Transcutaneous electrical nerve stimulation (TENS); Hyperbaric oxygen therapy (HBOT); Touch, Music & Aroma (TMA); Sleep, Music & Aroma (SMA); Repetitive transcranial magnetic stimulation (rTMS); Standard deviation (SD).

data (age, gender, ethnicity, diagnoses), intervention data (mechanism of action, description, frequency and duration, location, mode of delivery, follow-up), confounding variables, goal of intervention (management of condition or curative), subjective quantitative measures used, physiological measures used, qualitative measures used, data analysis method(s) used, findings, author conclusions and author limitations.

## Expert panel

Two meetings were held with the expert panel, who voluntarily contributed knowledge of lived experience of fatigue interventions as participants, carers and health and care professionals. An existing community network and the chronic fatigue service were approached to recruit to the panel along with those with academic interest and six participants attended. They offered initial context to confirm the search strategy and to inform the critical appraisal of findings. The group were actively engaged with and advised on the acceptability of interventions although were not expert in the review methods.

## Data analysis

**Narrative synthesis.** Narrative synthesis as described by Popay et al. [30] was used as the overarching framework of analysis and was used in combination with the statistical analysis to develop a textual summary account of the selected studies. This approach provided a non-linear framework comprising four elements enabling an iterative approach to analytical activities: developing a theory of how an intervention works and why; preliminary synthesis of findings; exploration of relationships; and assessing the robustness of the synthesis. The aim of narrative synthesis is not to reach a single conclusion or 'overall answer' to the research question, but rather to develop a trustworthy and compelling story by summarising the evidence, to present theoretical insights that aid and inform understanding of underlying mechanisms of action [30]. Synthesising the evidence in this way increases the chances of findings being used to inform real-world policy and practice.

First, an initial description of results from included studies and then a statistical analysis was conducted using a common rubric across all quantitative studies. This tabulation of results and enabled the examination of similarities and differences between outcomes and population groups according to fatigue severity of participants at baseline, intervention type, intervention duration and intensity and intervention effect size. Selecting interventions by virtue of reported effectiveness then enabled the comparison of components and characteristics. The findings were considered for areas of consensus and divergence, and a theory of outcome effectiveness for fatigue was developed. Finally, the robustness of the synthesis was assessed by critical reflection.

**Statistical analysis.** A standardised scale was developed to enable comparison of fatigue outcomes, which were assessed using 11 different fatigue measures across the included studies (see S2 File Comparison of fatigue measurement scales). The most used measure was the Chalder fatigue scale (n = 15). Any reported outcomes included dichotomous, or bimodal response scores were excluded from this analysis. Standardising the results was important to allow all available evidence to be included to avoid the risks associated with small overall sample sizes and wide confidence intervals. This method is valid where instruments measure the same or a similar construct [31].

Reported outcomes of fatigue interventions were converted to a common 0–100 scale, with 0 denoting no fatigue and 100 denoting the worst possible fatigue severity. Reported fatigue severity scores were thus expressed as a percentage of the maximum possible score for each given self-report fatigue scale, as reported in previous systematic reviews [32–34]. Changes in fatigue severity from baseline to post-intervention were calculated for both the control and experimental groups and the change in fatigue severity between baseline and follow-up was calculated. Mean differences in fatigue between the means of each experimental group and the control following the intervention in each study, as well as the 95% confidence intervals (CIs) for the mean difference.

Significance testing was conducted to assess statistical significance of differences in fatigue severity between groups following the intervention within each study. For studies where there was a control group and a single experimental group, independent sample t-tests were used; for studies where there was a control group and two or more experimental groups, one-way analysis of variance (ANOVA) was used to assess whether the fatigue severity of participants in one or more of the experimental groups was significantly different to the control group. If a significant difference was identified, Tukey's HSD post-hoc tests were conducted to determine which of the experimental groups differed significantly from the control. For both the t-test and the ANOVA analyses, a p-value of less than 0.05 was considered to be significant.

Standardised mean differences (SMD) were calculated as a secondary summary statistic to express the size effect of each intervention relative to the variability observed in that study. SMDs were calculated by subtracting the mean score of each intervention group from the mean score of the control group, divided by the pooled standard deviation of both groups at the post-intervention assessment [31]. Cohen's d is the appropriate effect size measure if both groups have similar standard deviations and are of the same size. Hedges' g, provides a measure of effect size weighted according to the relative size of each sample, and is an alternative where there are different sample sizes between groups. For both Cohen's d and Hedges' g, standardised mean difference cut-off points of 0.20, 0.50, and 0.80 can be considered to represent a small, moderate, and large effect, respectively [35]. All statistical analyses were conducted within SPSS (IBM SPSS Statistics 24.0).

**Content analysis.** Whilst the heterogeneity of intervention descriptions and outcome measures limited the ability to carry out a meta-analysis, the narrative synthesis enabled the further analysis of the characteristics of the interventions. We developed summary tables, to further consider the clinical context, intervention components, and direction of effect in relation to the participant experience of fatigue. Recognising the complexity of effect on psychological and functional wellbeing, it was possible to identify the complex descriptions of protocols and consider issues such as the training and competency associated with delivery and the recipient's personal context and commitment to sustained involvement, given the nature of their condition. All extracted data was used to identify a number of 'threads' [36] associated with the most effective interventions, based on the face validity to participants. This process enabled the synthesis of findings and a method to describe the rationale for managing fatigue and recommendations for the treatment of long Covid.

## Results

### Study selection

The search process yielded 13,613 results, which was reduced to 6,777 records after removal of duplicates. After review of abstracts, 115 papers were selected for full text screening, of which 40 were selected for inclusion in review. Papers were excluded at full text screening stage for reasons including being uncontrolled, not assessing fatigue as a primary outcome, not being published in English language, having only pharmacological components, having unclear analysis, and not having high validity. 37 out of the final 40 papers were deemed appropriate for further statistical analysis due to measures used (Fig 1).

### Study characteristics

Final selection yielded 40 studies reflecting an international body of literature spanning three continents, Europe, America, and Asia, plus one paper from Australia. Around 40% of the studies selected were published since 2016, while the remainder were published between 2003 and 2015. Publications include scientific reports and clinical trials in a range of psychological and physical health journals, including specific focus on Pain Management and Chronic Fatigue Syndrome (CFS). The fatigue interventions reported were associated with several post-viral syndromes, in 25 cases with CFS/ME, in nine cases for fibromyalgia and eight linked to unexplained fatigue. The mean participant age reported in individual studies ranged from 33.2 to 57.6 years, with women making up most of the participants (Table 1).

The selected studies reported a wide variety of approaches and treatments aimed at reducing fatigue. Broadly, these can be categorised as follows—although a few interventions straddle one or more categories. Firstly, those which are purely physiological in nature, such as transcranial magnetic [38]/transcutaneous electrical nerve [39] stimulation, hyperbaric oxygen

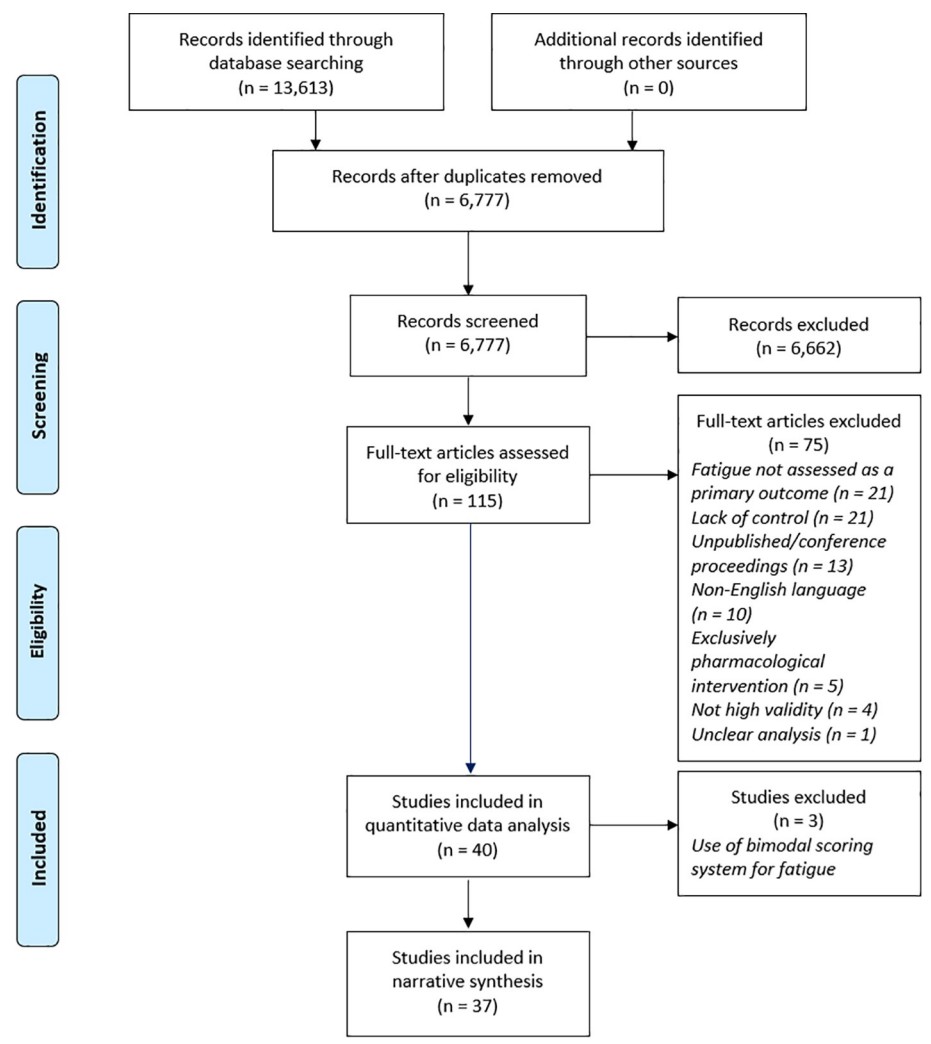

*From*: Moher, D., Liberati, A., Tetzlaff, J., Altman, G., The PRISMA Group (2009). Preferred reporting items for systematic reviews and meta-analyses: the PRISMA statement. BMJ, 339, b2535. doi: 10.1136/bmj.b2535 For more information, visit www.prisma-statement.org

**Fig 1. PRISMA flow diagram.** Adapted from Moher et al. [37].

therapy [40], homeopathy [41], acupuncture [42–46] and its variants–e.g. Tuina [47] and moxibustion [48]–all of which aim to stimulate or otherwise influence physiological processes (for example, increasing oxidation of the blood). Second, psychological or psycho-spiritual therapies such as cognitive behavioural therapy (CBT) [14, 49–53], behavioural self-regulation [54], operant learning [55] and Three Principles/Innate Health [56] which focus on influencing cognitive processes, building self-awareness and education. Third, physical interventions that require active physical input or participation from the individual, and which may or may not comprise a 'mind-body' component, such as Graded Exercise Therapy (GET) [57–62], Oriental Medicine Music Therapy [63], Tai Ji Quan [64] and yoga [65], and finally multi-modal interventions which comprise a either a mixture of the afore-mentioned activities or one or more activities plus pharmacological components [66–70]. Only one intervention focused solely on social support with no participation or activity undertaken by the fatigued individual

[71]. All interventions were controlled and typically compared with 'usual care' (a medical or multidisciplinary team general assessment and treatment) or a waiting list for the same.

In nine studies, the interventions were delivered solely in community settings such as general and specialist clinics [41, 43, 46, 47, 55, 58, 65, 70, 72], while a further nine comprised elements of home-based self-management combined with primary or outpatient care [53, 54, 61, 62, 64, 71, 73–75]. Two studies combined primary and secondary/specialist care [51, 52], while 11 studies combined secondary/specialist care with outpatient care and laboratory-based assessment [14, 38, 42, 49, 50, 57, 59, 60, 63, 67, 68]. Five studies were entirely laboratory-based [40, 44, 45, 48, 76], while four studies omitted details of treatment setting [39, 56, 66, 69]. In the majority of studies (28 of 40), treatment was administered or delivered by either clinically trained professionals or trained specialists (such as in the case of yoga and acupuncture). Four interventions were entirely or partly self-led by the participant at home, while in a small number of studies the intervention was delivered by the researchers themselves. One intervention was delivered by students with a background in psychology or social work [71]. Duration of interventions was highly variable, ranging from four days to 12 months, although the majority of studies adopted interventions lasting for 12 weeks or less.

## Effectiveness of interventions

The intervention that resulted in the largest mean difference in fatigue severity for participants compared to a control was the Three Principles/Innate Health (3P/IH) psycho-spiritual mental health education programme [56] (MD -39.0, 95% CI -51.8 to -26.2, SMD = 1.670) (Fig 2). Interventions employing group-based CBT were also found to result in large reductions in fatigue severity compared to a control (MD -27.5, 95% CI -33.7 to -21.3, SMD = 1.170 [50]; MD -23.7, 95% CI -29.2 to -18.2, SMD = 2.266 [51]).

A diverse range of interventions were found to result in significant reductions in fatigue severity compared to a control, including oriental medicine music therapy [63] (MD -21.7, 95% CI -30.4 to -12.9, SMD = 1.237); individual CBT [67] (MD -12.9, 95% CI -15.1 to -10.8, SMD = 1.784); yoga [65] (MD -20.0, 95% CI -34.6 to -5.4, SMD = 0.978); acupuncture [42] (MD -18.2, 95% CI -23.5 to -12.8, SMD = 0.984); and social support [71] (MD -12.0, 95% CI -20.7 to -3.4, SMD = 0.826).

Several studies employed interventions that did show benefits but with smaller effect level, such as operant learning [55] (MD -10.6, 95% CI -22.5 to 1.3, SMD = 0.520) and different forms of physical activity, such as GET [49] (MD -11.5, 95% CI -17.6 to -5.4, SMD = 0.524) and self-regulation based physical activity [61] (MD -9.9, 95% CI -16.3 to -3.4, SMD = 0.513).

Several interventions demonstrated either little or no benefit to individuals compared to standard care in terms of reducing fatigue severity, such as transcutaneous electrical nerve stimulation (TENS) [39] (MD -6.0, 95% CI -16.9 to 4.9, SMD = 0.239); Tai Ji Quan [64] (MD -0.4, 95% CI -9.6 to 8.9, SMD = 0.023) and supportive listening [75] (MD 1.06, 95% CI -1.4 to 3.6, SMD = -0.111). In addition, the efficacy of various forms of medication in reducing fatigue were investigated, however, were largely found to have little or no effect, for example: Mirtazapine [69] (MD -3.0, 95% CI -6.8 to 0.7, SMD = 0.447) and Doxycycline [67, 68] (MD 6.3, 95% CI 3.7 to 8.8, SMD = -0.752; MD 8.75, 95% CI 6.36 to 11.14, SMD = -1.120).

Only nine trials performed follow-up assessments of fatigue severity following the initial intervention (e.g. at 6–12 months following intervention) (See S3 File). Of these, only three interventions were found to have resulted in significant sustained reductions in fatigue; acupuncture [42] (MD -15.7, 95% CI -21.5 to -9.8, SMD = 0.773); self-management and web diaries [54] (MD -7.0, 95% CI -11.8 to -2.2, SMD = 0.514); and multidisciplinary rehabilitation [70] (MD -12.9, 95% CI -20.8 to -5.1, SMD = 0.454).

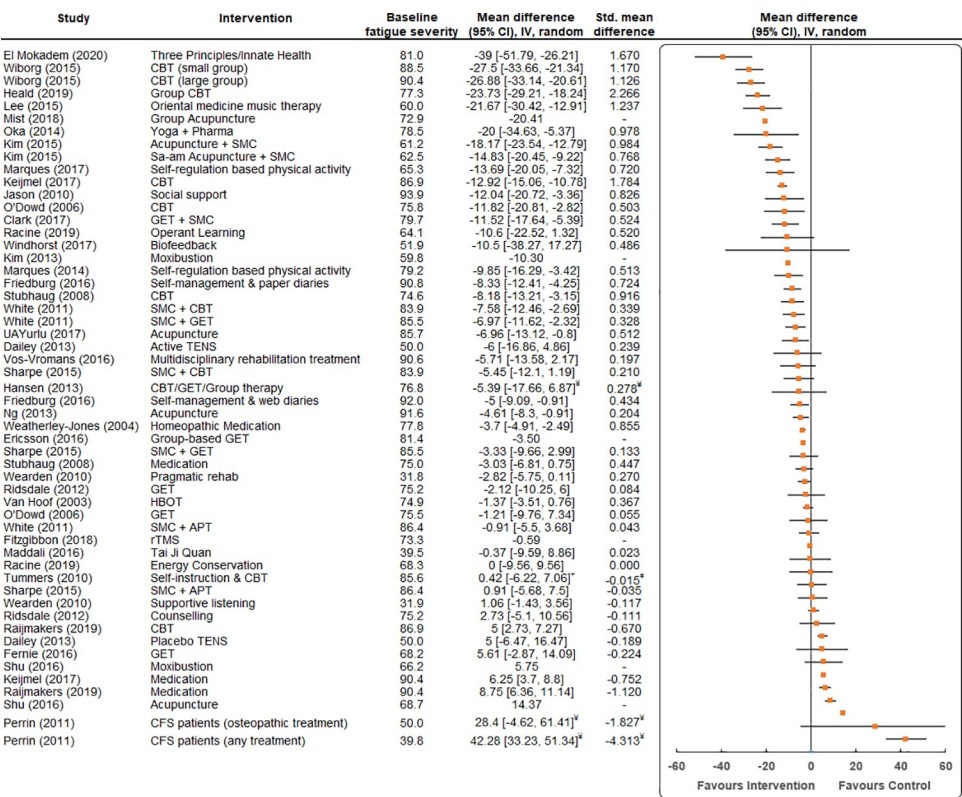

| Study | Intervention | Baseline fatigue severity | Mean difference (95% CI), IV, random | Std. mean difference |
| --- | --- | --- | --- | --- |
| El Mokadem (2020) | Three Principles/Innate Health | 81.0 | -39 [-51.79, -26.21] | 1.670 |
| Wiborg (2015) | CBT (small group) | 88.5 | -27.5 [-33.66, -21.34] | 1.170 |
| Wiborg (2015) | CBT (large group) | 90.4 | -26.88 [-33.14, -20.61] | 1.126 |
| Heald (2019) | Group CBT | 77.3 | -23.73 [-29.21, -18.24] | 2.266 |
| Lee (2015) | Oriental medicine music therapy | 60.0 | -21.67 [-30.42, -12.91] | 1.237 |
| Mist (2018) | Group Acupuncture | 72.9 | -20.41 | - |
| Oka (2014) | Yoga + Pharma | 78.5 | -20 [-34.63, -5.37] | 0.978 |
| Kim (2015) | Acupuncture + SMC | 61.2 | -18.17 [-23.54, -12.79] | 0.984 |
| Kim (2015) | Sa-am Acupuncture + SMC | 62.5 | -14.83 [-20.45, -9.22] | 0.768 |
| Marques (2017) | Self-regulation based physical activity | 65.3 | -13.69 [-20.05, -7.32] | 0.720 |
| Keijmel (2017) | CBT | 86.9 | -12.92 [-15.06, -10.78] | 1.784 |
| Jason (2010) | Social support | 93.9 | -12.04 [-20.72, -3.36] | 0.826 |
| O'Dowd (2006) | CBT | 75.8 | -11.82 [-20.81, -2.82] | 0.503 |
| Clark (2017) | GET + SMC | 79.7 | -11.52 [-17.64, -5.39] | 0.524 |
| Racine (2019) | Operant Learning | 64.1 | -10.6 [-22.52, 1.32] | 0.520 |
| Windhorst (2017) | Biofeedback | 51.9 | -10.5 [-38.27, 17.27] | 0.486 |
| Kim (2013) | Moxibustion | 59.8 | -10.30 | - |
| Marques (2014) | Self-regulation based physical activity | 79.2 | -9.85 [-16.29, -3.42] | 0.513 |
| Friedburg (2016) | Self-management & paper diaries | 90.8 | -8.33 [-12.41, -4.25] | 0.724 |
| Stubhaug (2008) | CBT | 74.6 | -8.18 [-13.21, -3.15] | 0.916 |
| White (2011) | SMC + CBT | 83.9 | -7.58 [-12.46, -2.69] | 0.339 |
| White (2011) | SMC + GET | 85.5 | -6.97 [-11.62, -2.32] | 0.328 |
| UAYurlu (2017) | Acupuncture | 85.7 | -6.96 [-13.12, -0.8] | 0.512 |
| Dailey (2013) | Active TENS | 50.0 | -6 [-16.86, 4.86] | 0.239 |
| Vos-Vromans (2016) | Multidisciplinary rehabilitation treatment | 90.6 | -5.71 [-13.58, 2.17] | 0.197 |
| Sharpe (2015) | SMC + CBT | 83.9 | -5.45 [-12.1, 1.19] | 0.210 |
| Hansen (2013) | CBT/GET/Group therapy | 76.8 | -5.39 [-17.66, 6.87]‡ | 0.278¥ |
| Friedburg (2016) | Self-management & web diaries | 92.0 | -5 [-9.09, -0.91] | 0.434 |
| Ng (2013) | Acupuncture | 91.6 | -4.61 [-8.3, -0.91] | 0.204 |
| Weatherley-Jones (2004) | Homeopathic Medication | 77.8 | -3.7 [-4.91, -2.49] | 0.855 |
| Ericsson (2016) | Group-based GET | 81.4 | -3.50 | - |
| Sharpe (2015) | SMC + GET | 85.5 | -3.33 [-9.66, 2.99] | 0.133 |
| Stubhaug (2008) | Medication | 75.0 | -3.03 [-6.81, 0.75] | 0.447 |
| Wearden (2010) | Pragmatic rehab | 31.8 | -2.82 [-5.75, 0.11] | 0.270 |
| Ridsdale (2012) | GET | 75.2 | -2.12 [-10.25, 6] | 0.084 |
| Van Hoof (2003) | HBOT | 74.9 | -1.37 [-3.51, 0.76] | 0.367 |
| O'Dowd (2006) | GET | 75.5 | -1.21 [-9.76, 7.34] | 0.055 |
| White (2011) | SMC + APT | 86.4 | -0.91 [-5.5, 3.68] | 0.043 |
| Fitzgibbon (2018) | rTMS | 73.3 | -0.59 | - |
| Maddali (2016) | Tai Ji Quan | 39.5 | -0.37 [-9.59, 8.86] | 0.023 |
| Racine (2019) | Energy Conservation | 68.3 | 0 [-9.56, 9.56] | 0.000 |
| Tummers (2010) | Self-instruction & CBT | 85.6 | 0.42 [-6.22, 7.06]* | -0.015* |
| Sharpe (2015) | SMC + APT | 86.4 | 0.91 [-5.68, 7.5] | -0.035 |
| Wearden (2010) | Supportive listening | 31.9 | 1.06 [-1.43, 3.56] | -0.117 |
| Ridsdale (2012) | Counselling | 75.2 | 2.73 [-5.1, 10.56] | -0.111 |
| Raijmakers (2019) | CBT | 86.9 | 5 [2.73, 7.27] | -0.670 |
| Dailey (2013) | Placebo TENS | 50.0 | 5 [-6.47, 16.47] | -0.189 |
| Fernie (2016) | GET | 68.2 | 5.61 [-2.87, 14.09] | -0.224 |
| Shu (2016) | Moxibustion | 66.2 | 5.75 | - |
| Keijmel (2017) | Medication | 90.4 | 6.25 [3.7, 8.8] | -0.752 |
| Raijmakers (2019) | Medication | 90.4 | 8.75 [6.36, 11.14] | -1.120 |
| Shu (2016) | Acupuncture | 68.7 | 14.37 | - |
| Perrin (2011) | CFS patients (osteopathic treatment) | 50.0 | 28.4 [-4.62, 61.41]¥ | -1.827¥ |
| Perrin (2011) | CFS patients (any treatment) | 39.8 | 42.28 [33.23, 51.34]¥ | -4.313¥ |

**Fig 2. Mean differences (95% Confidence Intervals (CI), Inverse Variance (IV)) for fatigue in studies assessing the effectiveness of interventions for reducing fatigue severity.** Fatigue is expressed on a 0–100 scale. Studies are ordered by the mean difference between the average score of participants in the intervention group and participants in the control group. The baseline fatigue severity of each group and the effect size (standardised mean difference (SMD, Cohen's d and Hedges' g)) of each intervention is also listed. ‡ = non-inferiority trial to see if one treatment is no less effective than the other, ¥ = fatigued individuals vs healthy controls.

## Characteristics of effective interventions

A diverse range of intervention approaches appear to have had a positive effect on severely fatigued individuals. A closer inspection of effective interventions reveals some complexity in terms of potentially uncontrolled elements (Table 2). For example, six out of 17 effective interventions comprised CBT delivered mostly as group interventions although in one case as a long series of traditional one to one session over 24 weeks [67]. Additional components such as peer support, structured or self-managed exercise programmes and body awareness therapy, provided additional value to participants insofar as social participation, peer support and joke-telling were specifically included. These elements may improve intervention acceptability, as has been acknowledged by at least one of the authors [51]. The effective CBT interventions in this review are inaccurately described as 'psychological therapies' given the additional social foci.

Three studies adopted self-regulation/behavioural self-management approaches with a focus on pacing, which appeared to be effective for those with relatively greater levels of fatigue severity, especially when supported by additional elements including Motivational Interviewing and counselling [61, 62]. Graded Exercise Therapy was shown to be effective only when adopted as a self-management strategy and supported by regular counselling sessions or specialist care.

**Table 2. Name of intervention with specific activities and components.**

| Author (Year) | Intervention | Intervention Activities/Components |
|---|---|---|
| Heald (2019) | CBT (Group) | Group CBT with joke-telling and peer support over 8 weeks |
| Keijmel (2017) | CBT | 24 weeks individual CBT |
| El Mokadem (2020) | Three Principles/Innate Health | Psycho-spiritual education comprising weekly educational videos, reading materials, webinars, individual coaching sessions, Facebook group over 8 weeks |
| Lee (2015) | Oriental Medicine Music Therapy | Relaxation, singing and music-making activities with traditional Korean instruments, 2–3 times/week over 2 weeks |
| Wiborg (2015) | CBT (Small Group) | CBT, patient feedback to group, 14 sessions over 6 months |
| | CBT (Large Group) | CBT, patient feedback to group, 14 sessions over 6 months |
| Kim (2015) | Acupuncture + SMC | 10 sessions of body acupuncture treatment for 4 weeks, 2 to 3 times/week |
| Oka (2014) | Isometric Yoga + Conventional Pharmacotherapy | 20-minute yoga session with instructor, 4 times over 2 months, home practice with DVD and booklet, conventional pharmacotherapy (details not stated) |
| Stubhaug (2008) | Comprehensive CBT | 2 x 1.5 hr sessions of group CBT/week, 1.5 hr body awareness therapy/week, self-managed exercise programme with exercise diary over 24 weeks |
| Weatherley-Jones (2004) | Homeopathic Medication | Classical homeopathic prescribing on a bespoke basis, monthly over 6 months |
| Jason (2010) | Social support | Support 'buddies' providing emotional support and functional support (such as household tasks), 2 hrs/week over 4 months |
| Kim (2015) | Sa-am Acupuncture + SMC | 10 sessions of Sa-am acupuncture and usual care for 4 weeks, 2–3 times/week |
| Friedberg (2016) | Behavioural self-management with paper diary | Pedometer use, answer daily questions via a paper diary to assess increases in activity or exercise, pacing of activities, greater exposure to pleasant activities, coping practices over 3 months |
| Marques (2017) | Self-regulation Based Physical Activity | 2 x motivational interviewing sessions, 2 x self-regulation-based telephone counselling sessions, information booklet, self-regulation-based workbook divided into 4 steps each one focusing on specific self-regulation cognitions and skills, pedometer use over 12 weeks |
| Clark (2017) | Guided graded exercise self-help plus specialist medical care | Six-step programme of graded exercise self-management based on the approach of GET, up to 3 support sessions by telephone/Skype |
| Marques (2014) | Self-regulation Based Physical Activity | 2 x motivational interviewing sessions, information booklet, self-regulation-based workbook divided into 4 steps each one focusing on specific self-regulation cognitions and skills, pedometer use over 12 weeks |
| O'Dowd (2006) | CBT (Group) | CBT, structured incremental exercise programme following group discussion, bi-weekly over 16 weeks |
| Friedberg (2016) | Behavioural self-management with web-based diary | Pedometer use, answer daily questions to assess increases in activity or exercise, pacing of activities, greater exposure to pleasant activities, coping practices over 3 months |
| White (2011) | CBT + SMC | CBT (fear avoidance theory), specialist medical care over 24 weeks |
| | GET + SMC | Graded exercise therapy (deconditioning and exercise intolerance theory), specialist medical care over 24 weeks |
| Ng (2013) | Acupuncture | Standard acupuncture (Traditional Chinese Medicine), 8 sessions over 4 weeks |

Standardised mean difference (SMD); Cognitive behavioural therapy (CBT); Chronic Fatigue Syndrome (CFS); Graded Exercise Therapy (GET); Randomised controlled trial (RCT); Standard Medical Care (SMC).

Two successful interventions adopted a 'mind-body' approach, one focusing on isometric yoga (with participants receiving pharmacological therapy) [65] and the other on psycho-spiritual education to improve body awareness and promote innate health [56]; while quite different in method and requirements of the participant, both appeared effective at symptom reduction in those with relatively higher levels of fatigue. Acupuncture and Oriental Medicine Music Therapy [63] showed positive effects in those with relatively lower levels of fatigue. Classical homeopathy, which arguably needed the least 'active input' by the participant, was shown to be effective in only one study. The study with the most severely fatigued individuals [71] adopted a very different approach to others, requiring nothing of the participant and focusing on allowing the fatigued individual to rest whilst providing functional and social support in

the guise of a visiting student to support with household tasks. This 'hands off' approach was shown to be highly effective at reducing fatigue in a severely fatigued sample group.

There does not appear to be any links between length of treatment time and effectiveness of intervention.

## Discussion

This study uses a systematic review process to identify evidence of effectiveness of interventions that reduce fatigue following viral infection. The authors sought to elicit learning that can be applied in practice, and which can be used to support service planning for fatigue management in the context of long Covid [77].

A vast range of treatment components, delivery modes and intervention durations were observed in this review. Despite significant levels of effectiveness reported in multiple studies, the range and complexity of intervention approaches adopted, and the homogeneity of participants, precludes a conclusive answer to the question, 'what works, and for whom'. Though international in scope, the current research evidence applies to a narrow range of people with fatigue, a relatively homogeneous group of patients in an age group between 45–55, with a mean age of 49 yrs. There is a considerable and growing realisation that a range of social and economic determinants may inform Covid-19 outcomes [78]; for example, lower family income and poor health status together with lack of physical activity, hypertension, and persistent Covid-19 symptoms are associated with recovery from fatigue-type symptoms [21]. A few key principles, however, can be drawn from the findings of this review, especially when considered in the context of the 'lived experience' of patients and practitioners, and the practical and operational considerations of wide-scale implementation with large numbers of patients with long Covid. In accordance with NICE guidance 2020, graded exercise is not indicated [79], but it seems that a range of activities and treatments can be offered in the context of individual choice and cultural context.

### The importance of person-centred planning

The severity of fatigue experienced by participants is a factor does not correlate with the type of intervention used but the real-world implementation is demonstrated in studies that use home-based interventions with remote or highly practical support for those with the most severe symptoms. The rationale for these treatments is the continuing engagement and optimisation of daily life activities [5]. Symptom management is therefore based upon individualised assessments and tailored to the patient's situation and the minimisation of exertion; energy conservation, which has been referred to as the "Envelope Theory" [71]. Where an intervention has focused entirely on social and functional support and has legitimised inactivity completely [71], it appears to have been effective in reducing symptoms for the most severely fatigued individuals. Where Wearden et al. [75] introduces exercise, the programme is devised collaboratively with the patient rather than prescribed based on exercise testing, as in graded exercise therapy. Generally, the patient-centred goals of treatment are not explicit in the literature, and yet those interventions that focus on individual experience and the functional impacts of fatigue on everyday life demonstrate some of the best outcomes.

### The importance of describing the participation required

Multi-modal approaches to interventions are under- reported overall or described as educational or technical interventions in UK, European and USA based studies. This is although many have social or emotional support elements to delivery that may contribute to effectiveness. This may reflect the demand from patients to address their physiological condition and

rather than focus on psychological components arising from fatigue. Yoga and acupuncture have no such dilemma and participants engage in treatment appreciating the holistic nature of the treatment activity that is meant to strengthen mind and body but are described in their cultural context where homeopathic interventions are the norm. Given the range of symptoms associated with long Covid and the known aetiology of the condition, it is perhaps important to recognise that patients expect allopathic interventions and may be very unused to needing psychological support. The early indications from the recent literature about rehabilitation services for long Covid indicate that the management of fatigue is a central focus and that an educational approach is being adopted incorporating energy management and rest [80], similar to those advocated by Friedberg [54] and Marques [61, 62].

## The importance of strengthening mind and body

According to traditional Oriental medicine, chronic fatigue is the result of an unbalanced state among inter-functioning organs, or a deficiency in vital energy, known as qi, and that by inserting acupuncture needles into specific points in the body this energy flow is brought back into proper balance. Yoga, while presented by a study in this review as a physical activity requiring active participation of the fatigued individual, is nonetheless recognised as a mind-body discipline that over time can contribute to changes in life perspective, increased self-awareness, and raised expectations regarding personal energy and life enjoyment [81]. Fatigue is both depressing and de-conditioning [82] and methods that enable people to engage with early rehabilitation are advocated [83] and it is becoming apparent that a broad range of accessible services are needed to offer rehabilitation as a means to prevent individuals from chronic illness [84].

## The importance of skilled and competent practitioners

Most interventions identified are based on high level competencies although there are those that do not require delivery by cognitive behavioural therapists or allied health professionals [71, 75]. The scale of the long Covid epidemic and numbers of people expected to need support, suggest that the effective interventions offering remote monitoring and support may be highly acceptable and enable access to those from more marginal communities who are not yet accessing long Covid services. Interventions offered in the context of community rehabilitation need to be based on their scalability and real-world practicality, as well as efficacy. Whilst early consensus advocates for holistic care, investigation of specific symptom clusters and individualised rehabilitation [85] there is a need to recognise the potential to consider awareness raising and fatigue management across community services and the wider social care workforce.

## The importance of observational research

The fatigue interventional studies included here, reflect a highly diverse range of methods which are assessed for effectiveness and some which can be replicated in further studies with people who have long Covid. There are some group interventions that appear to work well and others offering minimal, remote interventions that could be scaled to meet the probable demand. The remaining interventions may not be culturally acceptable or are not available to be prescribed via health and social care provision. There are methods requiring careful observational studies with clinical academic designs, scaled to provide guidance for learning in a practice context. Systematic process evaluation is required to ensure that the academic fraternity uses public and patient involvement to understand the acceptability of complex

evaluations that are based on fatigue experience in the long Covid population. Rapid evaluation of the acceptable activities and health outcomes are urgently needed.

## Strengths and limitations

The review is carried out to a high quality and includes systematic mixed methods of synthesising the findings and critically evaluating their useful in the context of long Covid fatigue management. In the final selection only papers adopting a quantitative methodology were included and very limited qualitative or mixed methods data was available. It focuses on the critical appraisal of the research evidence in relation to the mechanism of the interventions and on the measurement of effectiveness. These two elements of analysis were deemed to be of priority interest when seeking to build a rationale for planning further fatigue interventions and looking for a robust theory to underpin interventions for fatigue in long Covid. It is a strength of this paper that we have undertaken an inclusive analysis across a range of treatment modalities, allowing in the future for a range of novel treatments to be considered and analysed. In the additional appraisal of on-going clinical studies there were no further interventional studies that needed to be included in this review but a wide variety of studies into idiopathic chronic fatigue and also some that focussed on the needs of children and young people, outside the scope of this investigation.

The authors acknowledge the limitations of the narrative synthesis process used to assess and summarise the findings within this review. Quality of evidence is varied, and in developing a narrative to describe and explain the varied results, assumptions have been made to fill gaps and manage uncertainties. For example, the lack of focus in most studies on the functional abilities and baseline severity of fatigued individuals means that effectiveness cannot fully be considered on a 'like-for-like' basis, and this may call into question the overall validity and generalisability of the findings. In tabulating the characteristics of effective interventions alongside baseline fatigue severity in this review, an attempt has been made to limit assumptions about the condition of participating individuals; nonetheless, the lack of information regarding the functional capabilities of the study participants *before* they participated in the intervention was a key topic of discussion during the synthesis process.

The findings suggest a critical need for further research that includes a diversity of ages and socio-economic groups, and improved reporting of demographic and cultural variation in the management of fatigue. long Covid may affect 10% of the UK population but a far higher number of people who had Covid were from the worst deprived communities [86] suggesting that long Covid will similarly be distributed and may lead to significant inequalities of service provision. There are no studies that target the assessment and experience of fatigue and the management of fatigue with socially and economically deprived people.

## Conclusion

The aim of the study was to systematically review the literature associated with fatigue management interventions, their characteristics, and outcomes, and to identify treatments that may be useful in the management of long Covid. NICE guidelines were published [87], but the evidence relating to fatigue management treatments for long Covid were limited. The findings of this study suggest that the fatigue management research is highly focussed on a narrow participant demographic, of which 18 interventions demonstrated no effect in managing fatigue symptoms. The selected literature, demonstrates how self-rated fatigue scales are used to report the effect of complex interventions, which is a limitation of the body of research as a whole. Synthesis suggests that long Covid fatigue management may be beneficial when: a) physical and psychological support, is delivered in groups where people can plan their functional

response to fatigue [27]; and b) where strengthening rather than endurance is used to prevent deconditioning; and c) where fatigue is regarded in the context of an individual's lifestyle and home-based activities are used. Further research is required into fatigue management with more diverse populations.

## Supporting information

**S1 Checklist. PRISMA 2020 checklist.**
(DOCX)

**S1 File. Search strategy.** Table 1. MEDLINE Search; Table 2. ProQuest Search.
(DOCX)

**S2 File. Comparison of fatigue measurement scales.**
(DOCX)

**S3 File. Overview of intervention effectiveness using standardised measures, including follow-up assessments.**
(DOCX)

## Acknowledgments

The authors thank all the members of an expert advisory group who met twice during the study and who shared their experience of fatigue, service access and treatment and commented on the previous research into fatigue, also commenting on scope, initial findings and early research outcomes.

## Author Contributions

**Conceptualization:** Sally Fowler-Davis, Amie Woodward.

**Data curation:** Sally Fowler-Davis, Michael Thelwell, Amie Woodward.

**Formal analysis:** Sally Fowler-Davis.

**Funding acquisition:** Sally Fowler-Davis.

**Investigation:** Sally Fowler-Davis, Katharine Platts, Deborah Harrop.

**Methodology:** Sally Fowler-Davis, Katharine Platts, Michael Thelwell, Amie Woodward, Deborah Harrop.

**Project administration:** Sally Fowler-Davis, Katharine Platts, Michael Thelwell, Amie Woodward.

**Supervision:** Deborah Harrop.

**Validation:** Deborah Harrop.

**Visualization:** Michael Thelwell.

**Writing – original draft:** Sally Fowler-Davis, Katharine Platts, Michael Thelwell.

**Writing – review & editing:** Sally Fowler-Davis, Katharine Platts, Michael Thelwell, Amie Woodward, Deborah Harrop.

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
