## [Decision Letter · Decision Letter 0]

11 Aug 2021

PONE-D-21-18781

A mixed-methods systematic review of post-viral fatigue interventions: are there lessons for long Covid?

PLOS ONE

Dear Dr. Fowler Davis,

Thank you for submitting your manuscript to PLOS ONE. After careful consideration, we feel that it has merit but does not fully meet PLOS ONE’s publication criteria as it currently stands. Therefore, we invite you to submit a revised version of the manuscript that addresses the points raised during the review process.

We look forward to receiving your revised manuscript.

Kind regards,

Sinan Kardeş, M.D.

Academic Editor

PLOS ONE

Journal Requirements:

Reviewers' comments:

Reviewer's Responses to Questions

**Comments to the Author**

1. Is the manuscript technically sound, and do the data support the conclusions?

Reviewer #1: Yes

Reviewer #2: Partly

Reviewer #3: Partly

2. Has the statistical analysis been performed appropriately and rigorously? 

Reviewer #1: Yes

Reviewer #2: Yes

Reviewer #3: I Don't Know

3. Have the authors made all data underlying the findings in their manuscript fully available?

Reviewer #1: Yes

Reviewer #2: Yes

Reviewer #3: Yes

4. Is the manuscript presented in an intelligible fashion and written in standard English?

Reviewer #1: Yes

Reviewer #2: Yes

Reviewer #3: Yes

5. Review Comments to the Author

Reviewer #1: The aim of this study is to synthesise the literature associated with fatigue interventions to investigate the outcomes that may be applicable to ‘Long Covid’. The article reports an innovative and interesting work.

In the introduction I suggest to add reference to the extent of the long covid syndrome, the correct definition (with further subdivisions depending on whether the symptoms develop before or after four weeks, according to NICE guidelines) and hints of the physiopathology justifying the different treatments.

The aims of the study are well explained. The structure of the study is well organized. The materials and methods sections are discussed in details. I suggest to add in the text the initials of the author who screened the articles. Statistical analysis is well conducted and appropriate.

The results are clearly explained in the text and there are not repetitions between the text and the table. However, I recommend making the table clearer and faster to compare. At the end of each table I would insert the abbreviations used.

In addition, the methods explain that the selected studies were related to post-viral fatigue, in the tables, however, studies related to fibromyalgia symptoms are shown. Please clarify this important concern.

In the discussion section I suggest to add reference to Maccarone MC, Magro G, Tognolo L, Masiero S. Post COVID-19 persistent fatigue: a proposal for rehabilitative interventions in the spa setting [published online ahead of print, 2021 Jun 4]. Int J Biometeorol. 2021;1-3. doi:10.1007/s00484-021-02158-1.

The work weakness has already been identified by the authors in the limitations section. In addition, making an inclusive analysis of methods of medicine recognized by the scientific community, alternative medicine methods, psychological and mind-body interventions can be a limitation because it risks confusing the reader or suggesting solutions not considered to be used at first. It would be appropriate to discuss this point further to avoid confusion.

Reviewer #2: Dear Authors,

It has been a pleasure to review your article and the topic is of great importance in the field of rehabilitation medicine. However, in my opinion, there are some issues that should be addressed.

1. I think, that in the aim of the study should be used “systematically” instead of “systemic”

2. Under Eligibility criteria it is said that “mixed interventions and studies involving a pharmacological component were included only when fatigue outcome measurement data could be extracted. It is not clear whether pharmacological interventions, if they were not mixed with other methods, were included or not. Were studies reporting other interventions, but not reporting fatigue outcome measurement were included? This is confusing. Could you please clarify?

3. Could you, please, clarify the 10% of articles that was screened by the second reviewer? The reasoning behind this approach is not obvious in the text.

4. Results of the quality appraisal of included studies could be reported in Table 1.

5. Could you more precisely describe the expert panel and their role in the study? How many were they? Which professions? Why they can be called experts? How their input affected decisions and results?

6. Since the effect sizes has been calculated based on the outcome measures used in the included studies, it would be important to include these measures in the descriptions of studies (Table 1 or Figure2).

7. Since the information overlaps between Table 2 and Figure 2, consider omitting Table 2 from the article.

8. Although, discussion includes several interesting and important aspects of fatigue treatment, it lacks the summary and interpretation of the direct results of the study.

9. Conclusions should contain only information that directly answers to research question(s) included in the aim of the study and are based on the results of this study. Please, revise conclusions.

Reviewer #3: The authors performed a study addressing “A mixed-methods systematic review of post-viral fatigue interventions: are there lessons for long Covid?”. There is one point that needs to be corrected.

I think the Conclusion is too long. Can you make it concise? It is unclear what you want to show in this research paper.

6. PLOS authors have the option to publish the peer review history of their article (what does this mean?). If published, this will include your full peer review and any attached files.

Reviewer #1: No

Reviewer #2: **Yes: **Guna Berzina

Reviewer #3: No

---

## [Author Response · Author response to Decision Letter 0]

16 Sep 2021

Response to Reviewers

Additional journal requirements.

The literature search strategy is in Supplementary Information S1_File. The supporting information includes three additional files, which are submitted along with the manuscript. The work does not include any primary data.

Reviewer #1: 

The aim of this study is to synthesise the literature associated with fatigue interventions to investigate the outcomes that may be applicable to ‘Long Covid’. The article reports an innovative and interesting work.

Thank you for these comments we appreciate your consideration of our work.

In the introduction I suggest to add reference to the extent of the long covid syndrome, the correct definition (with further subdivisions depending on whether the symptoms develop before or after four weeks, according to NICE guidelines) and hints of the physiopathology justifying the different treatments.

The references 17-21 include the current literature associated with Long Covid syndrome especially in ref 20, the specific range of international thinking associated with the treatment choices. We think the addition of the NICE guideline is out of place given the focus on fatigue management.

The aims of the study are well explained. The structure of the study is well organized. The materials and methods sections are discussed in details. I suggest to add in the text the initials of the author who screened the articles. 

We have amended this according to your suggestion. 

Statistical analysis is well conducted and appropriate. 

Thank you

The results are clearly explained in the text and there are not repetitions between the text and the table. However, I recommend making the table clearer and faster to compare. At the end of each table I would insert the abbreviations used.

We have amended this according to your suggestion. 

In addition, the methods explain that the selected studies were related to post-viral fatigue, in the tables, however, studies related to fibromyalgia symptoms are shown. Please clarify this important concern.

We have taken Fibromyalgia to be a post-viral syndrome see www.healthguideinfo.com/fibromyalgia/p113865/ that highlights the Epstein-Barr virus that causes a range of immunodeficiency. The identification of fibromyalgia is widely accepted to our knowledge and understanding.

In the discussion section I suggest to add reference to Maccarone MC, Magro G, Tognolo L, Masiero S. Post COVID-19 persistent fatigue: a proposal for rehabilitative interventions in the spa setting [published online ahead of print, 2021 Jun 4]. Int J Biometeorol. 2021;1-3. doi:10.1007/s00484-021-02158-1.

This reference provides a compelling and interesting treatment option for post viral fatigue but does not appear to be a good fit in the conclusion, but we will be citing in another article related to treatment modalities in sport and leisure facilities.

The work weakness has already been identified by the authors in the limitations section. In addition, making an inclusive analysis of methods of medicine recognized by the scientific community, alternative medicine methods, psychological and mind-body interventions can be a limitation because it risks confusing the reader or suggesting solutions not considered to be used at first. It would be appropriate to discuss this point further to avoid confusion.

Thanks, we have added one sentence to reflect this concern in the analysis but we see this as a strength of the paper P26

Reviewer #2: 

Dear Authors, it has been a pleasure to review your article and the topic is of great importance in the field of rehabilitation medicine. However, in my opinion, there are some issues that should be addressed.

Thanks for your review, we have made changes as follows

1. I think, that in the aim of the study should be used “systematically” instead of “systemic”

Thank you, this is corrected on page 5.

2. Under Eligibility criteria it is said that “mixed interventions and studies involving a pharmacological component were included only when fatigue outcome measurement data could be extracted. It is not clear whether pharmacological interventions, if they were not mixed with other methods, were included or not. Were studies reporting other interventions, but not reporting fatigue outcome measurement were included? This is confusing. Could you please clarify?

We have added a statement to clarify on page 5 to say that pharma studies were excluded.

3. Could you, please, clarify the 10% of articles that was screened by the second reviewer? The reasoning behind this approach is not obvious in the text.

We have added the following 10% of the papers screened by a second review author to check for conformity of selection and to quality assess the level of agreement between reviewers

4. Results of the quality appraisal of included studies could be reported in Table 1. 

We have now included MMAT quality appraisal scores in Table 1 and have also provided comments about the preliminary use of the MMAT on page 7.

5. Could you more precisely describe the expert panel and their role in the study? How many were they? Which professions? Why they can be called experts? How their input affected decisions and results?

A short additional paragraph has been added page 8.

6. Since the effect sizes has been calculated based on the outcome measures used in the included studies, it would be important to include these measures in the descriptions of studies (Table 1 or Figure2).

Effect size values have been added to Figure 2.

7. Since the information overlaps between Table 2 and Figure 2, consider omitting Table 2 from the article.

Baseline severity and effect size values have been removed from Table 3 (which has also been correctly relabelled as table 2).

8. Although, discussion includes several interesting and important aspects of fatigue treatment, it lacks the summary and interpretation of the direct results of the study.

The summary is in the results section and an additional sentence is in the discussion

9. Conclusions should contain only information that directly answers to research question(s) included in the aim of the study and are based on the results of this study. Please, revise conclusions.

Conclusion is revised in line with your comments.

Reviewer #3: 

The authors performed a study addressing “A mixed-methods systematic review of post-viral fatigue interventions: are there lessons for long Covid?”. There is one point that needs to be corrected.

I think the Conclusion is too long. Can you make it concise? It is unclear what you want to show in this research paper.

This has been revised in line with point 9 of Reviewer 2.

---

## [Decision Letter · Decision Letter 1]

4 Oct 2021

PONE-D-21-18781R1A mixed-methods systematic review of post-viral fatigue interventions: are there lessons for long Covid?PLOS ONE

Dear Dr. Fowler Davis,

Thank you for submitting your manuscript to PLOS ONE. After careful consideration, we feel that it has merit but does not fully meet PLOS ONE’s publication criteria as it currently stands. Therefore, we invite you to submit a revised version of the manuscript that addresses the points raised during the review process. Please submit your revised manuscript by Nov 18 2021 11:59PM. If you will need more time than this to complete your revisions, please reply to this message or contact the journal office at plosone@plos.org. Please include the following items when submitting your revised manuscript:A rebuttal letter that responds to each point raised by the academic editor and reviewer(s). You should upload this letter as a separate file labeled 'Response to Reviewers'.A marked-up copy of your manuscript that highlights changes made to the original version. You should upload this as a separate file labeled 'Revised Manuscript with Track Changes'.An unmarked version of your revised paper without tracked changes. You should upload this as a separate file labeled 'Manuscript'.If applicable, we recommend that you deposit your laboratory protocols in protocols.io to enhance the reproducibility of your results. Protocols.io assigns your protocol its own identifier (DOI) so that it can be cited independently in the future. For instructions see: https://journals.plos.org/plosone/s/submission-guidelines#loc-laboratory-protocols. Additionally, PLOS ONE offers an option for publishing peer-reviewed Lab Protocol articles, which describe protocols hosted on protocols.io. Read more information on sharing protocols at https://plos.org/protocols?utm_medium=editorial-email&utm_source=authorletters&utm_campaign=protocols.

We look forward to receiving your revised manuscript.

Kind regards,

Sinan Kardeş, M.D.

Academic Editor

PLOS ONE

Journal Requirements:

Reviewers' comments:

Reviewer's Responses to Questions

**Comments to the Author**

1. If the authors have adequately addressed your comments raised in a previous round of review and you feel that this manuscript is now acceptable for publication, you may indicate that here to bypass the “Comments to the Author” section, enter your conflict of interest statement in the “Confidential to Editor” section, and submit your "Accept" recommendation.

Reviewer #2: (No Response)

2. Is the manuscript technically sound, and do the data support the conclusions?

Reviewer #2: Yes

3. Has the statistical analysis been performed appropriately and rigorously? 

Reviewer #2: Yes

4. Have the authors made all data underlying the findings in their manuscript fully available?

Reviewer #2: Yes

5. Is the manuscript presented in an intelligible fashion and written in standard English?

Reviewer #2: Yes

6. Review Comments to the Author

Reviewer #2: Dear Authors,

There are some more issues that I think you should clarify:

1. You have included quality appraisal using the MMAT. The scores and interpretation should be mentioned. And since it is score, I don’t think that * is needed in the table 1.

2. Please revise the title of Table 2.

3. You report fatigue severity scores in a common metric scale. I still think that it would be good to address the fact that the fatigue was measured using different scores – how many different scales were there, which were the most commonly used?

7. PLOS authors have the option to publish the peer review history of their article (what does this mean?). If published, this will include your full peer review and any attached files.

Reviewer #2: **Yes: **Guna Berzina

---

## [Author Response · Author response to Decision Letter 1]

12 Oct 2021

Thanks for the final clarification you requested;

1. You have included quality appraisal using the MMAT. The scores and interpretation should be mentioned. And since it is score, I don’t think that * is needed in the table 1.

All asterisks have been removed

 2. Please revise the title of Table 2.

Thanks, we have amended the title of the table

3 You report fatigue severity scores in a common metric scale. I still think that it would be to address the fact that the fatigue was measured using different scores – how many different scales were there, which were the most commonly used?

We have included a supplementary file 2 that addresses the range of measures and this is referred to on line 219 of page 9 but we have added a small amount of detail to identify the most commonly used in the following section

Finally

We noted that the abstract and the conclusion were not quite the same and so we have slightly amended the abstract to ensure the same message

---

## [Decision Letter · Decision Letter 2]

21 Oct 2021

A mixed-methods systematic review of post-viral fatigue interventions: are there lessons for long Covid?

PONE-D-21-18781R2

Dear Dr. Fowler Davis,

We’re pleased to inform you that your manuscript has been judged scientifically suitable for publication and will be formally accepted for publication once it meets all outstanding technical requirements.

Kind regards,

Sinan Kardeş, M.D.

Academic Editor

PLOS ONE

Reviewers' comments:

Reviewer's Responses to Questions

**Comments to the Author**

1. If the authors have adequately addressed your comments raised in a previous round of review and you feel that this manuscript is now acceptable for publication, you may indicate that here to bypass the “Comments to the Author” section, enter your conflict of interest statement in the “Confidential to Editor” section, and submit your "Accept" recommendation.

Reviewer #2: All comments have been addressed

2. Is the manuscript technically sound, and do the data support the conclusions?

Reviewer #2: Yes

3. Has the statistical analysis been performed appropriately and rigorously? 

Reviewer #2: Yes

4. Have the authors made all data underlying the findings in their manuscript fully available?

Reviewer #2: Yes

5. Is the manuscript presented in an intelligible fashion and written in standard English?

Reviewer #2: Yes

6. Review Comments to the Author

Reviewer #2: (No Response)

7. PLOS authors have the option to publish the peer review history of their article (what does this mean?). If published, this will include your full peer review and any attached files.

Reviewer #2: **Yes: **Guna Berzina

---

## [Editor Report · Acceptance letter]

29 Oct 2021

PONE-D-21-18781R2 

A mixed-methods systematic review of post-viral fatigue interventions: are there lessons for long Covid? 

Dear Dr. Fowler Davis:

I'm pleased to inform you that your manuscript has been deemed suitable for publication in PLOS ONE. Congratulations! Your manuscript is now with our production department. 

Kind regards, 

on behalf of

Dr. Sinan Kardeş 

Academic Editor

PLOS ONE